# Modeling of Isometric Muscle Properties via Controllable Nonlinear Spring and Hybrid Model of Proprioceptive Receptors

**DOI:** 10.3390/muscles4030029

**Published:** 2025-08-11

**Authors:** Mario Spirito

**Affiliations:** Italian Institute of Technology (IIT), Via Morego, 30, 16163 Genoa, Italy; mario.spirito@iit.it

**Keywords:** skeletal muscles isometric force modeling, nonlinear variable spring, hybrid dynamical modeling, Spindle-Ia & -II, oculomotor system

## Abstract

This work investigates the macroscopic behavior of skeletal muscles from a system-theoretic perspective. Based on data available in the literature, we propose an initial evaluation model for isometric force generation, i.e., force produced at a constant muscle length or in quasi-static conditions, as a function of muscle length and neuronal excitation frequency. This model enables a more physics-inspired representation of isometric force by employing a nonlinear spring framework with controllable properties such as stiffness and rest length. Finally, we introduce a hybrid dynamical filter model to describe components of the sensory system responsible for relaying information about muscle length and its rate of change back to the Central Nervous System. As an application case, we present the modeling of the oculomotor system, highlighting the relevance of the proposed modeling approach in a physiologically meaningful control task.

## 1. Introduction

Skeletal muscles, or simply muscles, are organs of the muscular system whose cells have the ability to generate force and movement in response to electrical stimulation. They are primarily attached to the bones of the skeleton via tendons and, because of their contractile and elastic properties, enable the relative motion of bones at the joints they actuate.

Each skeletal muscle consists of thousands of muscle fibers bundled together and enclosed by connective tissue sheaths (see, e.g., [1], Chap. 9). These bundles of muscle fibers are called fasciculi. Each muscle fiber contains numerous myofibrils, which in turn are made up of repeating units of myofilaments. When assembled, these myofibrils exhibit a characteristic striated pattern, forming sarcomeres, the fundamental contractile units of skeletal muscle. The two primary types of myofilaments are actin (thin) and myosin (thick), which are arranged in a regular pattern that gives rise to the distinct banding appearance of skeletal muscle [2,3,4] (Chap. 1).

Skeletal muscles appear with neuronal innervations of the sensory-motor system. A set of neurons, i.e., the α-motoneurons, constitutes the neuromuscular junction. They are responsible, through their excitation frequency (Here labeled as α, measured in pulses per second, or pps), for the sarcomeres contractions, and hence they allow the generation of the active force of the muscle. On the other hand, another set of neurons, which constitute proprioceptive receptors, carries sensorial information from the muscle to the Central Nervous System. In particular, transmitted signals are the values of muscle length and rate of change (Spindles-Ia and -II) and the measurement of the force value at the tendon level (Spindle-Ib or Golgi Tendon Organs (GTO)). We can see the muscle structure from a system-theoretic standpoint as a multi-input multi-output system with the actuator (α-motoneurons), the plant (the muscle cells, whose contraction produces the exerted force on the joint), and the sensors (Spindles -Ia and -II, and GTO), and the CNS as a feedback controller. A schematic of the neurological connections involved is depicted in Figure 1, inspired by [5].

The modeling of how muscles exert force started with the work of [6], in which muscle description is presented through a spring-like behavior. In particular, the relationship takes into account a contractile element that produces muscle force and it is put in series to a passive elastic element representing the tendon model between the muscle fibers and the connected bones or tissues. This model has then been exploited to generate more detailed models [7,8] in which the authors describe the microscopic behavior of contraction behavior generated by the effects of sarcomeres and cross-bridges. In [9], Hill’s model has been enriched with a ‘damping element’. In [10,11,12], experimental data have been used to construct an analytic expression of the total force generated by muscles. In these works, the total force has been divided into a length-dependent component $F_{L}$ and a velocity-dependent component $F_{V}$, both modulated by the excitation frequency of the α-motoneuron. The $F_{L}$ term is further decomposed, in line with Hill’s model, into an active and a passive component. The active force accounts for the contraction generated by muscle fiber activation and is responsible for movement, while the passive force reflects the intrinsic elastic response of muscle fibers in the absence of activation. A fundamental reference in this context is the work of Zajac [13], which provides a comprehensive overview of muscle and tendon dynamics and has laid the groundwork for many modern neuromechanical models. Experimental data and force generation models are available for isometric (fixed muscle length) and isokinetic (fixed muscle velocity) configurations and have been discussed in [10,11,12,14,15].

In [9,16,17], a system-theoretic perspective on the neuromuscular system has been proposed, where the authors provide a control-oriented interpretation of the various physiological components involved. In [18], the focus is instead on the optimal co-activation problem, with the aim of minimizing the mechanical impedance of a biological joint actuated by antagonistic muscles (e.g., the elbow).

In recent years, the contractile properties of the neuromuscular system, particularly those involving Hill’s model equations, have been further investigated in [19]. In [20], an analytical model of complete and incomplete tetanic contraction has been proposed which, to the author’s knowledge, represents the most realistic dynamical model of muscle behavior currently available.

The sensory components of the muscle system are responsible for measuring plant variables such as tendon force (through the Golgi Tendon Organs, GTOs) and muscle fiber length and its rate of change (through Spindles Ia and II). These elements have been extensively studied in [5,21], and a comprehensive review of related works is provided in [22]. For a more recent overview, see [23].

From a systems engineering perspective, a dynamical model of GTO behavior has been presented in [24], which introduces a transfer function that maps the tendon force to the firing frequency of GTO neurons. For Spindles Ia and II, a static model was proposed in [25] to describe the relationship between fiber length, its rate of change (velocity), and the excitation from γ-motoneurons to the spindle firing rate. These γ-motoneurons are responsible for modulating the response of the associated spindle system. The sensory system plays a crucial role in the architecture of movement control. In particular, the presence of internal loops between the muscles and the spinal cord enables the activation of reflexes such as the stretch reflex, reciprocal inhibition, and others [17,26,27]. A key component of this architecture is the spinal cord, which connects the Central Nervous System (CNS) to the α-motoneurons via interneurons. It is inevitably involved in motor control processes [28,29,30].

The paper is organized as follows. We conclude this Introduction by providing the main motivations for the study. In Section 2, we discuss some considerations regarding the available experimental data. Section 3 presents the muscle models found in the literature, which have been derived on the basis of these data. In Section 4, we introduce the models developed for isometric muscle force. Section 5 is devoted to the hybrid modeling of the sensory system embedded in skeletal muscles. An application of the proposed modeling framework to the oculomotor system is presented in Section 6, along with a discussion of the related challenges. Finally, Section 7 concludes the paper by summarizing the main contributions and describing the directions for future research.

### Motivations

This work aims to describe neuromuscular behavior within a control-theoretic framework. In particular, we seek to gain a deeper understanding of the role of neuromuscular feedback loops and how they are exploited to control the movements of the limb. For example, in [31], Iqbal et al. show the application of a stabilizing PID controller to a single-link biomechanical model, while Kistemaker et al. illustrate how sensory feedback can be used effectively in position and movement control [32]. For insights into how waveform signals can be used to control delayed closed-loop systems, mimicking the spiking behavior of biological neural systems, see [19]. We believe that adopting this modeling perspective provides a promising starting point for gaining deeper insights into the roles of different brain regions involved in learning and controlling dexterous movements. For references on this broader neurological context, see [33,34,35,36,37,38]. This line of research could also support the modeling and simulation of pathological conditions affecting motor control, a long-term research goal.

To begin addressing this long-term objective, we discuss how the proposed nonlinear spring model may serve to capture altered muscle properties seen in pathological conditions, such as increased stiffness in spasticity or reduced passive force in muscle atrophy. Furthermore, we suggest that the hybrid spindle model could, with appropriate parameter tuning, reproduce abnormal stretch reflex behavior, for example, exaggerated reflex sensitivity in hyperreflexia. To improve the specificity of the analysis, future extensions of the model could include spinal interneuron circuits, which play a fundamental role in modulating reflex responses through inhibitory and excitatory pathways. Accurate modeling of these components may help distinguish whether abnormal motor behavior arises from changes in peripheral sensors (e.g., muscle spindles), central processing (e.g., interneuronal gain) or efferent pathways, thus offering a more precise understanding of the underlying pathophysiology. In addition to structural or neurological impairments, it is important to recognize that factors such as emotional state, fatigue, and muscle temperature can also significantly affect muscle behavior. These influences can alter both the stiffness parameters and the resting length of muscle fibers, leading to variability in force generation and reflex response. Therefore, incorporating such modulatory effects, possibly through time-varying or state-dependent parameters, may enhance the physiological realism and clinical applicability of the proposed models.

Although this study does not include clinical data, the structure of the models is designed to be extendable to such scenarios. In Section 7, we outline how future work could incorporate clinical datasets to refine and validate the pathological relevance of these models. The mid-term goal is to develop a comprehensive understanding of how the brain organizes and controls human dexterous motion, with the aim of replicating such strategies in the design of advanced nonlinear control systems, such as the nonlinear output regulation problem addressed in [39]. This study intersects with the concepts of inverse and forward models, as introduced in neuroscience in [40,41]. See also [42] for further applications of these principles. An intermediate step in this process, in our view, involves improving our understanding of how these models are formed, trained, and updated in the brain and how they contribute to the overall control strategy.

## 2. Preliminaries on Available Experimental Data in the Literature

In the muscle force, we can distinguish three main components: a passive force $F_{p}$ that is always present due to the natural behavior of tissues and depends only on muscle length; an active static (isometric) force $F_{L}$ that depends solely on muscle contraction length *L*; and an active dynamic force (eccentric and concentric) $F_{V}$ which also depends on the rate of change of muscle contraction $\dot{L}$. The last two components are active terms and depend on the activation frequency α of the α-motoneuron. The isometric force data available in the literature are mainly those shown in the figures of [11,12,18], which are summarized in this section for ease of reference.

It is worth noting that these data cover the activation frequency only within the interval [0, 120] pps (pulses per second), as 120pps is the maximum excitation frequency that the α-motoneuron can ‘apply’ to the muscle fiber. At α=120 pps, the active force $F_{L}$ (i.e., the total force excluding the passive $F_{p}$ and dynamic $F_{V}$ components) reaches a maximum value, denoted in the literature by the symbol $F_{o}$. This maximum force $F_{o}$ corresponds to a specific muscle length $L_{o}$ (which is not the maximum length, but the length at which the force $F_{o}$ is reached at α=120 pps). Almost all available data and models in the literature are normalized with respect to these two values; therefore, force and length data are typically presented as $F/F_{o}$ and $L/L_{o}$, respectively. Accordingly, we consider a normalized model based on these normalized quantities.

A crucial aspect in our model development is the presence of rest-length values in the force characteristic (that is, values of *L* for which the exerted force is zero) that depend on the excitation frequency α (see [18]). This leads us to introduce a rest length (here, by rest length, we mean the value of the muscle length for which, at a given excitation frequency, the exerted force is zero). function $L_{0}$ (distinct from $L_{o}$) that depends solely on the excitation frequency α, which can be used to describe muscle contraction behavior. A significant challenge is that most of the data in the literature lack information on these rest-length values, which are essential for our model development and motivated the development of two models of muscle behavior, as described in the following sections. To address this, we adopt the technique introduced in [11]. Specifically, they first proposed a model without rest length, based on Gaussian terms (also detailed in Section 3), and then added a negative passive term to compensate for the absence of rest length data. This adjustment allows the rest length points to saturate the force at zero, as clearly shown in [11].

In the next section, we provide an overview of two prominent models available in the literature.

## 3. Comparison of Literature Models

We consider the models presented in [11,12] to be the most relevant and comprehensive. A brief description of these models is given here for completeness.

### 3.1. Brown et al. Model

In [12], similarly to Hill’s classical model [6], the authors represent both active isometric and dynamic force generation by a contractile element $F_{CE}$, while the passive force is described by a passive elastic element $F_{PE}$ (previously denoted as $F_{p}$, but here we retain the original notation). The total force $F_{TOT}$ is given by(1)$$F_{TOT}=F_{CE}+F_{PE}$$
where the contractile element force is modeled as(2)$$F_{CE}\left(\alpha ,L,\dot{L}\right)=R\left(\alpha \right)\cdot A\left(\alpha ,L\right)\cdot F_{L}\left(L\right)\cdot F_{V}\left(L,\dot{L}\right)$$
where *R* represents the percentage of fiber recruitment and quantifies the proportion of fibers that contribute to force generation depending on the activation frequency α. The term *A* relates muscle activation to excitation frequency, taking values in the interval [0, 1] (the same range applies to *R*). The functions $F_{L}$ and $F_{V}$ describe the force-length and force-velocity relationships, respectively. Their product provides muscle force as a function of current length *L*, contraction velocity $\dot{L}$, and excitation frequency α.

The authors also present a simplified force model by fixing R(α)=1 and assuming quasi-static (nearly isometric) conditions during motion, with the muscle starting at rest length. This leads to the expression:(3)$$F_{CE}\left(\alpha ,L,\dot{L}\right)=A_{f}\left(\alpha ,L\right)\cdot \left(F_{L}\left(L\right)\cdot F_{V}\left(L,\dot{L}\right)+F_{p}\left(L\right)\right)$$
where(4)$$\begin{array}{cc} A_{f}\left(\alpha ,L\right) & =1-\exp\left(-\frac{\alpha }{0.56\cdot N_{f}\left(L\right)}\right)\\ N_{f}\left(L\right) & =2.11+4.16\left(\frac{1}{L}-1\right)\\ F_{L}\left(L\right) & =\exp\left(-{\left|\frac{L^{1.93}-1}{1.03}\right|}^{1.87}\right)\\ F_{p}\left(L\right) & =-0.02\exp(13.8-18.7L)\\ F_{V}\left(L,\dot{L}\right) & =\left\{\begin{array}{cc} \frac{-5.72-\dot{L}}{-5.72+\left(1.38+2.09L\right)\cdot \dot{L}}, & \dot{L}\leq 0\\ \frac{2.5+4.21L+2.67L^{2}}{0.62+\dot{L}}\dot{L}, & \dot{L}>0 \end{array}\right. \end{array}$$

It should be noted that in this model, the force length $F_{L}$ and force velocity $F_{V}$ components are multiplied, which can be interpreted as the velocity term $F_{V}$ acting as a scaling factor on the isometric force $F_{L}$. We prefer to treat the isometric $F_{L}$ and dynamic $F_{V}$ terms separately to analyze their individual contributions in a spring-damper analogy.

Moreover, observe that as $\dot{L}\to 0^{+}$, the velocity term $F_{V}\neq 1$, indicating a loss of continuity with the static isometric muscle properties following a lengthening phase.

### 3.2. Winters’ Model

In [11], Winters’ work delves deeply into the basic muscle components. Specifically, the study provides a microscopic description of muscle elements such as intrafusal and extrafusal fibers, along with their associated damping and stiffness terms, connected in both parallel and series arrangements. In this section, we focus exclusively on the isometric force, modeled as$$F_{ISO}\left(\alpha ,L_{CE}\right)=F_{L}\left(\alpha ,L_{CE}\right)+F_{p}\left(L_{CE}\right)$$
where the active isometric force $F_{L}$ is described by a Gaussian function with a variable mean:(5)$$F_{L}\left(\alpha ,L_{CE}\right)=\alpha \exp\left(-{\left(\frac{\frac{L_{CE}}{L_{m0}}-1.05-\left(1-\alpha \right)\cdot 0.2}{0.4}\right)}^{2}\right)$$
Here, $L_{m0}$ corresponds to the optimal muscle length $L_{o}$ in this work’s notation, while$$L_{CE}=L_{mt}-L_{t0}-x_{SE}\left(L_{m0}+L_{t0}\right)$$
with $x_{SE}$ representing the dimensionless extension of the series of extrafusal elastic elements, $L_{mt}$ the total musculotendinous length, and $L_{t0}$ the rest length of the tendon. The activation level α is normalized relative to its maximum.

The passive force is modeled as(6)$$F_{p}=\left\{\begin{array}{cc} \frac{\exp\left(\frac{3}{0.6}\cdot \left(\frac{L_{CE}}{L_{m0}}-1\right)\right)-1}{e^{3}-1}, & \frac{L_{CE}}{L_{m0}}\geq 1\\ -\frac{\exp\left(\frac{6}{0.7}\cdot \left(1-\frac{L_{CE}}{L_{m0}}\right)\right)-1}{e^{6}-1}, & \frac{L_{CE}}{L_{m0}}<1 \end{array}\right.$$Note that, as mentioned earlier, the passive force is negative for $\frac{L_{CE}}{L_{m0}}<1$.

We believe that for the purposes of this study, the functions used in the models described here are quite complex to manipulate and apply in control analysis. Furthermore, the physical interpretation of their parameters is challenging.

## 4. Proposed Muscle Models

Our main objective is to describe the isometric force $\left(F_{L}+F_{p}\right)$ as a nonlinear controlled spring with variable stiffness *K* and rest length $L_{0}$, that is,(7)$$F_{ISO}\left(\alpha ,L\right)=K\left(\alpha ,L\right)\cdot \left(L-L_{0}\left(\alpha \right)\right)$$
where α is the excitation frequency of the α-motoneuron, and *L* is the normalized muscle length (with respect to $L_{o}$, the length corresponding to the optimal-maximal-muscle force at a given excitation frequency α).

To clarify, we provide precise definitions of rest length and optimal length:

The *rest length* $L_{0}\left(\alpha \right)$ is defined as the muscle length *L* at which, for a given excitation frequency α, the muscle generates no force, i.e.,$$F_{ISO}\left(\alpha ,L_{0}\left(\alpha \right)\right)=0.$$
The *optimal length* $L_{o}\left(\alpha \right)$ is defined as the muscle length *L* at which the isometric force is maximized for the given excitation frequency α, that is,$$\frac{\partial F_{ISO}}{\partial L}\left(\alpha ,L_{o}\left(\alpha \right)\right)=0.$$

To obtain this final model, we first designed an intermediate evaluation model based on Winters’ considerations. In particular, we exploited the negative passive term introduced in [11] to determine the rest length values for different excitation frequencies. This step enables us to later construct and fit the parameters of the final control model (Equation 7).

### 4.1. Evaluation Model

To build the intermediate evaluation model, we consider the isometric force as composed of active $F_{L}$ and passive $F_{p}$ components, as in [11]:$$F_{ISO}\left(\alpha ,L\right)=F_{L}\left(\alpha ,L\right)+F_{p}\left(L\right).$$
Inspired by the modeling approach in [12], we describe active force curves as Gaussian functions with mean value μ, standard deviation σ, and amplitude *A*, all depending on the excitation frequency α, as shown in (Equation 8). In this context, the mean value parameter μ corresponds to the optimal muscle length $L_{o}$, the coefficient *A* represents the maximal force generated at a given excitation frequency α, and the standard deviation σ characterizes the width of the force–length relationship around the optimal length, reflecting the mechanical properties of the muscle at that activation level.(8)$$\begin{array}{cc} F_{L} & =A\exp\left(-{\left(\frac{L-\mu }{\sigma }\right)}^{2}\right)\\ A & =\frac{c_{11}\alpha^{2}+c_{12}\alpha +c_{13}}{\alpha^{2}+c_{14}\alpha +c_{15}}\\ \mu & =c_{21}\exp\left(c_{22}\alpha \right)+c_{23}\exp\left(c_{24}\alpha \right)\\ \sigma & =c_{31}+c_{32}\cos\left(c_{34}\alpha \right)+c_{33}\sin\left(c_{34}\alpha \right) \end{array}$$

By running a curve fitting algorithm, we were able to fit the active isometric force with a very good approximation; specifically, the standard Euclidean norm of the residual vector is ${\left|r\right|}_{2}=0.051$, where *r* represents the vector of differences between the available force data and the corresponding values predicted by the proposed model. In Figure 2, we plot a comparison of the active force between the data from [12] and the intermediate model estimation. The identified coefficients are reported in Table 1. Using the data available in [11] and their considerations on negative passive terms, we propose a unified continuous model of the passive force term, exploiting a sigmoidal and exponential function:(9)$$F_{p}\left(L\right)=f_{1}+\frac{f_{2}-f_{1}}{1+10^{f_{3}\cdot \left(f_{4}-L\right)}}+f_{5}\exp\left(f_{6}L\right)$$
where the coefficient values $f_{i}$, for i={1,…,6}, can be found in Table 2.

The sigmoidal behavior in the passive term (see Figure 3) does not have a direct correspondence with experimental data but provides a plausible description of the physical limit of any elastic material; that is, beyond the ‘boundary’ length value, the solicited fibers begin to break.

The total isometric force is then given by saturating all negative values at zero, expressed as(10)$$F_{ISO}\left(\alpha ,L\right)=\left\{\begin{array}{cc} F_{p}\left(L\right)+F_{L}\left(\alpha ,L\right), & \mathrm{if}\;F_{ISO}\left(\alpha ,L\right)>0,\\ 0, & \mathrm{otherwise}. \end{array}\right.$$

### 4.2. Control Model

Due to the high complexity of the intermediate/evaluation model (Equation 10), and to make the force relationship more intuitive from a physical perspective, we used the intermediate model to extract the rest length points and corresponding equivalent stiffness values. These data were essential to fit the parameters of the rest length $L_{0}\left(\alpha \right)$ and stiffness K(α,L) functions in terms of α and *L*, thus constructing the final (control) model. The isometric force is thus given by (Equation 7), which we report again here for completeness:(11)$$F_{ISO}\left(\alpha ,L\right)=K\left(\alpha ,L\right)\cdot \left(L-L_{0}\left(\alpha \right)\right)$$
where both $L_{0}$ and *K* are modeled as polynomial functions in α and *L*:(12)$$\begin{array}{cc} L_{0}\left(\alpha \right) & =\ell_{1}\alpha^{2}+\ell_{2}\alpha +\ell_{3},\\ K(\alpha ,L) & =k_{1}\left(L\right)\alpha^{3}+k_{2}\left(L\right)\alpha^{2}+k_{3}\left(L\right)\alpha +k_{4}\left(L\right), \end{array}$$
where each $k_{i}\left(L\right)$, for i=1,…,4, is itself a polynomial function in *L*:(13)$$k_{i}\left(L\right)=k_{i1}L^{6}+k_{i2}L^{5}+k_{i3}L^{4}+k_{i4}L^{3}+k_{i5}L^{2}+k_{i6}L+k_{i7}.$$

The corresponding coefficients for the rest length and stiffness functions are reported in Table 3 and Table 4, respectively.

We also report in Figure 4 a comparison between the stiffness surface fitted K(α,L) and the corresponding data points extracted from the Evaluation Model, for varying values of α and *L*.

In Figure 5, we show the comparison between the Control Model and the experimental data. The residual vector *r* represents the difference between the model output and the available data.

The model provides a good approximation of the Evaluation Model and, consequently, of the experimental data, as shown in Figure 5. Moreover, the Control Model offers a reliable reconstruction of the total isometric force within the range $\left[0.4L_{o},\;1.5L_{o}\right]$, as illustrated in Figure 6. In particular, the residual norm is relatively small, with ${\left|r\right|}_{2}=0.18114$, especially for $L<1.5L_{o}$.

Since muscle lengths beyond $1.5L_{o}$ are generally outside the range of motion allowed by biological joints due to mechanical constraints, the proposed control model is well suited for applications involving standard physiological movements.

## 5. Sensors Modeling

In this section, we discuss the dynamic modeling of the sensory organs involved in muscle feedback. These sensors measure muscle length, contraction velocity, and force applied at the tendon level. Specifically, the primary sensors are the Spindle types II and Ia, which encode muscle length and velocity, and the type Ib afferents, also known as Golgi Tendon Organs (GTOs), which measure tendon force. For a detailed model of the Golgi Tendon Organs, we refer the reader to the work in [24,43].

### 5.1. Spindles Ia and II

The neuronal activity of the spinal afferents Spindle-Ia and Spindle-II is correlated with both the length of the muscle fiber and its rate of change. Each Spindle-Is innervated by active elements, specifically γ-motoneurons, whose excitation frequency modulates the spindle’s sensitivity. In particular, this modulation can be interpreted as an enhancement of sensing precision, albeit at the cost of increased noise amplification. We can generally distinguish between static ($\gamma_{s}$) and dynamic ($\gamma_{d}$) γ-motoneurons. In the present work, we focus on the role of the dynamic input of the γ-motoneuron ($\gamma_{d}$) in modulating the spindle response, since the available data include activation patterns for $\gamma_{d}$ only. Although static γ-motoneurons ($\gamma_{s}$) also influence spindle sensitivity, their effect is not modeled here due to data limitations. This assumption may affect the completeness of the feedback dynamics captured by the model and will be addressed in future developments.

As shown in [22] (Chap. 19), the Spindle-Ia output saturates at zero during negative velocity transients, while for Spindle-II the behavior is less clear, probably due to the termination of $\gamma_{s}$ excitation during the same phase of the motion trajectory. To construct sensor models, we first analyze the static behavior of sensor outputs and estimate their slopes, following an approach similar to that in [25]. We now present the hybrid dynamical model developed for Spindle-Ia, followed by the model for Spindle-II.

#### 5.1.1. Spindle-Ia

We modeled the Spindle-Ia sensor using a hybrid dynamical system, defined as follows:(14)$$\begin{array}{cc} C_{Ia} & :\left\{\begin{array}{c} {\dot{x}}_{Ia}=p_{Ia}\cdot \left(K_{L_{Ia}}L-x_{Ia}\right)\\ {\dot{p}}_{Ia}=0\\ {\dot{K}}_{L_{Ia}}=0\\ y_{Ia}={\mathrm{sat}}_{0}\left(x_{Ia}+K_{\gamma_{Ia}}\gamma_{Ia}+K_{{\dot{L}}_{Ia}}\dot{L}\right) \end{array}\right.\\ D_{Ia} & :\left\{\begin{array}{c} \begin{array}{cccc} \; & \left(\forall X_{Ia}\in D_{Ia1}\right) & \;K_{L_{Ia}}^{+}=0,\; & p_{Ia}^{+}={\bar{p}}_{Ia}\\ \; & \left(\forall X_{Ia}\in D_{Ia2}\right) & \;K_{L_{Ia}}^{+}={\bar{K}}_{L_{Ia}},\; & p_{Ia}^{+}={\bar{p}}_{Ia}\\ \; & \left(\forall X_{Ia}\in D_{Ia3}\right) & \;K_{L_{Ia}}^{+}=0,\; & p_{Ia}^{+}={\underline{p}}_{Ia}\\ \; & \left(\forall X_{Ia}\in D_{Ia}\right) & \;x_{Ia}^{+}=x_{Ia},\; & y_{Ia}^{+}=y_{Ia} \end{array} \end{array}\right. \end{array}$$

Here, $x_{Ia}$ is the filter state and $y_{Ia}$ is the model output. The function ${\mathrm{sat}}_{0}\left(\cdot \right)$ denotes saturation at zero for negative values. The parameter $p_{Ia}$ is the filter eigenvalue, and $K_{L_{Ia}}$, $K_{{\dot{L}}_{Ia}}$, and $K_{\gamma_{Ia}}$ are static gains associated with muscle length, velocity, and dynamic excitation of γ motor neurons, respectively. The signals *L*, $\dot{L}$, and $\gamma_{Ia}$ represent muscle length, its time derivative, and the input of the $\gamma_{d}$ -motoneuron, which are treated as input of the system.

The state vector is defined as $X_{Ia}:={\left(x_{Ia},\dot{L},\gamma_{Ia}\right)}^{⊤}$, and the hybrid system is defined in the flow set $C_{Ia}=R^{3}\setminus D_{Ia}$ and the jump set $D_{Ia}=D_{Ia1}\cup D_{Ia2}\cup D_{Ia3}$, where:$$\begin{array}{cc} D_{Ia1} & =\left\{X_{Ia}\in R^{3}:\dot{L}<0,\;\left(K_{L_{Ia}}\neq 0\;\vee \;p_{Ia}\neq {\bar{p}}_{Ia}\right)\right\}\\ D_{Ia2} & =\left\{X_{Ia}\in R^{3}:\dot{L}>0,\;\left(K_{L_{Ia}}\neq {\bar{K}}_{L_{Ia}}\;\vee \;p_{Ia}\neq {\bar{p}}_{Ia}\right)\right\}\\ D_{Ia3} & =\left\{X_{Ia}\in R^{3}:\dot{L}=0,\;\left(K_{L_{Ia}}\neq 0\;\vee \;p_{Ia}\neq {\underline{p}}_{Ia}\right)\right\} \end{array}$$

The identified parameters are listed in Table 5. The simulation result is plotted in Figure 7.

#### 5.1.2. Spindle-II

In addition, the Spindle-II sensing system is modeled using a hybrid dynamical system defined as:(15)$$\begin{array}{cc} C_{II} & :\left\{\begin{array}{c} {\dot{x}}_{II}=p_{II}\cdot \left(\xi_{L_{II}}K_{L_{II}}L-x_{II}\right)\\ {\dot{p}}_{II}=0\\ {\dot{\xi }}_{L_{II}}=0\\ y_{II}={\mathrm{sat}}_{0}\left(x_{II}+K_{\gamma_{II}}\gamma_{II}+K_{{\dot{L}}_{II}}\dot{L}+\left(1-\xi_{L_{II}}\right)\cdot K_{L_{II}}\cdot \left(L-\bar{L}\right)\right) \end{array}\right.\\ D_{II} & :\left\{\begin{array}{c} \begin{array}{cccc} \; & \left(\forall X_{II}\in D_{II1}\right) & \;\xi_{L_{II}}^{+}=0,\; & p_{II}^{+}={\bar{p}}_{II}\\ \; & \left(\forall X_{II}\in D_{II2}\right) & \;\xi_{L_{II}}^{+}=1,\; & p_{II}^{+}={\bar{p}}_{II}\\ \; & \left(\forall X_{II}\in D_{II3}\right) & \;\xi_{L_{II}}^{+}=0,\; & p_{II}^{+}={\underline{p}}_{II}\\ \; & \left(\forall X_{II}\in D_{II}\right) & \;x_{II}^{+}=x_{II},\; & y_{II}^{+}=y_{II} \end{array} \end{array}\right. \end{array}$$

Here, $x_{II}$ is the filter state and $y_{II}$ is the system output. The function ${\mathrm{sat}}_{0}\left(\cdot \right)$ represents a saturation function that sets negative values to zero. The parameters $p_{II}$, $K_{L_{II}}$, $K_{{\dot{L}}_{II}}$, and $K_{\gamma_{II}}$ are, respectively, the filter eigenvalue and static gains for muscle length, velocity, and excitation of the $\gamma_{d}$-motoneuron.

The inputs of the system are *L*, $\dot{L}$, and $\gamma_{II}$, denoting the muscle length, velocity, and $\gamma_{d}$-motoneuron excitation level. The hybrid state vector is defined as $X_{II}:={\left(x_{II},\dot{L},\gamma_{II}\right)}^{⊤}$. The flow set is given by $C_{II}=R^{3}\setminus D_{II}$, while the jump set is defined as $D_{II}=D_{II1}\cup D_{II2}\cup D_{II3}$, with:$$\begin{array}{cc} D_{II1} & =\left\{X_{II}\in R^{3}:\dot{L}<0,\;\left(\xi_{L_{II}}\neq 0\;\vee \;p_{II}\neq {\bar{p}}_{II}\right)\right\}\\ D_{II2} & =\left\{X_{II}\in R^{3}:\dot{L}>0,\;\left(\xi_{L_{II}}\neq 1\;\vee \;p_{II}\neq {\bar{p}}_{II}\right)\right\}\\ D_{II3} & =\left\{X_{II}\in R^{3}:\dot{L}=0,\;\left(\xi_{L_{II}}\neq 0\;\vee \;p_{II}\neq {\underline{p}}_{II}\right)\right\} \end{array}$$

The identified parameters are summarized in Table 6. The simulation result is plotted in Figure 8.

## 6. Application to the Oculomotor System

In this section, we apply the proposed muscle modeling framework to the oculomotor system. This application is motivated by the works [44,45,46], in which the oculomotor system is described under the assumption that actuation is not generated by muscular torque, but is rather directly modeled as an acceleration input, see Equation (11b) in [46]. In our opinion, this assumption oversimplifies the problem, suggesting that a more comprehensive analysis is required when considering an adaptive control formulation of the system.

We propose that a more realistic model must account for the activation of skeletal muscles through the firing rates of α-motoneurons, denoted $\alpha_{i}$, i=1,2. These activations generate muscle forces, which, in turn, produce a torque on the eyeball. According to the labeling in Figure 9, the horizontal motion of the eyeball is modeled as:(16)$$\begin{array}{cc} \dot{\theta } & =\omega \\ J\dot{\omega } & =-\beta \omega +R\cdot K_{1}\left(\alpha_{1},L_{10}-R\theta \right)\cdot \left(L_{10}-R\theta -L_{01}\left(\alpha_{1}\right)\right)\\ \; & \;-R\cdot K_{2}\left(\alpha_{2},L_{20}+R\theta \right)\cdot \left(L_{20}+R\theta -L_{02}\left(\alpha_{2}\right)\right) \end{array}$$
where θ is the angular displacement with respect to the sagittal plane, ω is its time derivative, *J* is the eyeball inertia, β is a viscous damping coefficient, and *R* is the eyeball radius. The quantities $\alpha_{i}$, $K_{i}$, $L_{i}$, and $L_{0i}$, for i=1,2, represent the α-motoneuron firing rates, muscle stiffness, instantaneous muscle lengths and rest lengths, respectively.

Furthermore, the length of each muscle has been written as function of the angular displacement θ, i.e., $L_{i}=L_{i0}+{\left(-1\right)}^{i}R\theta$, where $L_{i0}$ is the initial (neutral) length of the muscle *i*.

This model reveals that the assumption made in [46], that the torque can be directly controlled through acceleration, is no longer valid without making additional assumptions on the generation of muscle force. Specifically, a central nervous structure would need to provide an inverse mapping from the desired torque to the required firing rates $\alpha_{1}$ and $\alpha_{2}$. However, such an inverse map does not exist in general, since the forward map from the firing rates to the torque$$\tau =R\cdot K_{1}\left(\alpha_{1},L_{1}\right)\cdot \left(L_{1}-L_{01}\left(\alpha_{1}\right)\right)-R\cdot K_{2}\left(\alpha_{2},L_{2}\right)\cdot \left(L_{2}-L_{02}\left(\alpha_{2}\right)\right)$$
is surjective, but not injective. In particular, this redundancy can be exploited to achieve a desired joint stiffness independently of the torque, providing an additional degree of control.

Furthermore, the oculomotor system incorporates sensory feedback, as described in Section 5, with one sensory system per muscle. Assuming $L_{10}=L_{20}$, the scaled difference between $L_{2}$ and $L_{1}$ gives the angle θ, while the difference between ${\dot{L}}_{2}$ and ${\dot{L}}_{1}$ gives its rate of change. These signals can be reconstructed from the outputs of spindle types Ia and II in each muscle.

This sensory information operates in synergy with visual input from the cortex. For example, during object tracking, an additional error signal is provided that completes the oculomotor system.

### Simulation Study: Application to the Oculomotor System

To demonstrate the practical relevance of the proposed muscle and spindle models, we performed numerical simulations focused on the oculomotor system. The model parameters were set as reported in Table 7. We consider normalized mechanical parameters where *J* and *R* are adimensional unitary variables, i.e., J=1 and R=1.

Figure 10 shows the response to simulated muscle force during a typical saccadic eye movement, illustrating the interaction between the controllable nonlinear spring behavior and the spindle feedback dynamics. This simulation confirms that the proposed models can reproduce key dynamic features observed in oculomotor muscles and provides a foundation for further closed-loop control analysis.

## 7. Conclusions and Future Works

In this work, we deal with modeling muscle force considering the passive and active components of the isometric force. We first provided a brief overview of the model available in the literature and then showed two models, i.e., an intermediate model and a final model. The first is more complex and precise than the second, but it does not add much to the models available in the literature. On the other hand, it allowed us to construct a set of data that have been useful for the identification of the second model parameters. The latter (the spring model) is easier to handle, although less precise than the intermediate one, and gives a physical intuition about the muscle behavior. In fact, it describes such muscular behavior as a controllable nonlinear spring with controllable (variable) stiffness *K* and rest length $L_{0}$. We furthermore provide a hybrid dynamical model formulation of the spindle organs that act as muscle length and velocity sensors for the Central Nervous System. We then provide an application to the case of the oculomotor system proposing its dynamic modeling and describing the related challenges.

### 
Future Works


Future works may concern the development of a muscle force characteristic depending also on the rate of shortening and lengthening of the muscle fibers’ length, to provide a more complete model. This model can then be analyzed in a control context involving (biological) joints with antagonist actuation. Moreover, a system-theoretic description of physiological reflexes, such as the stretch reflex, can be investigated by exploiting the proposed model.

While the present study focuses on healthy neuromuscular behavior, the proposed models have been designed to allow extension to pathological conditions. Future work will aim to fit these models to clinical datasets—such as EMG and passive stretch data from patients with spasticity or neuromuscular disorders—to assess their ability to capture altered reflex dynamics and changes in muscle stiffness. This would provide a quantitative framework to differentiate between healthy and pathological motor control, with potential applications in diagnosis and rehabilitation. In addition, the proposed control-oriented framework may be extended to incorporate synergy-based motor primitives, following the approach of [47], and reflex modulation strategies such as those discussed in [48]. Furthermore, a quantitative comparison with computational muscle models used in biomechanics, such as the Millard muscle model [49] available in OpenSim, could help validate the proposed framework and facilitate its integration into existing simulation platforms. Although the current work focuses on a single-muscle model, future research will explore scalability to multi-muscle systems. This initial simplification was chosen to enable a tractable closed-loop control analysis, particularly in the presence of neural delays that naturally lead to infinite-dimensional system dynamics. To our knowledge, no formal analysis currently exists for such feedback-delayed neuromuscular systems. Therefore, this study sets the theoretical foundation for subsequent investigations that will consider more complex and physiologically accurate scenarios. Despite the relatively high number of parameters, the proposed model is entirely based on polynomial functions, offering fixed computational complexity and numerical efficiency that make it suitable for potential real-time applications. We acknowledge that the current model is limited to isometric conditions and does not capture force–velocity effects arising during muscle shortening or lengthening. While this simplification was instrumental in developing and validating the static force components, it naturally restricts the range of applications. Future extensions will incorporate dynamic muscle behavior to allow the modeling of concentric and eccentric contractions, enabling broader applicability to realistic motor tasks. Future work will also aim to validate the proposed models using datasets from different muscle groups and species, in order to assess their generalizability and robustness across diverse physiological contexts.

## Figures and Tables

**Figure 1 muscles-04-00029-f001:**
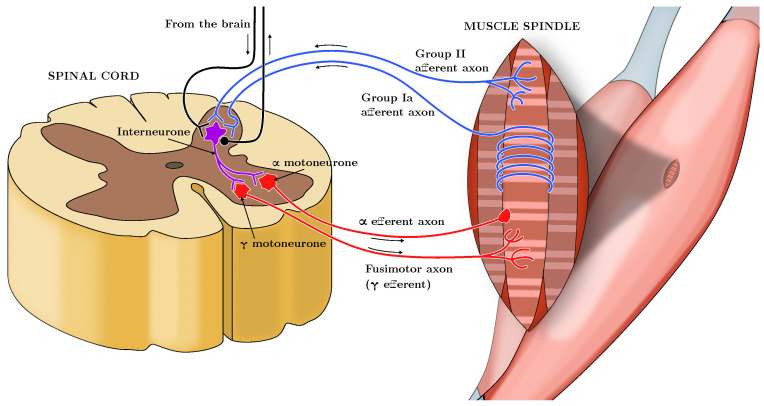
Schematic representation of the neuromuscular system, showing the α-motoneurons driving muscle contraction, the proprioceptive feedback from Spindles Ia, II, and Golgi Tendon Organs (GTO), and the role of the Central Nervous System (CNS) as a feedback controller. Inspired by [5].

**Figure 2 muscles-04-00029-f002:**
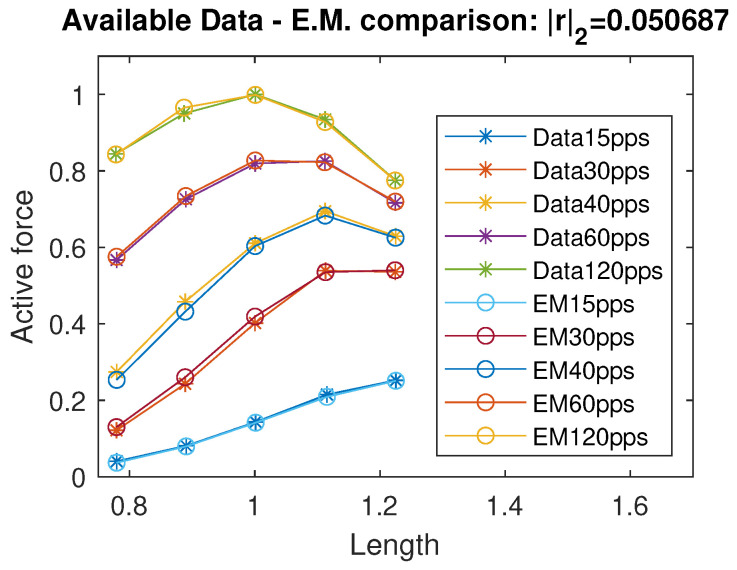
Data comparison between the available data in [12] and the evaluation/intermediate model. *r* indicates the residuals vector. All quantities are normalized with respect to the maximal active force $F_{o}$ and optimal length $L_{o}$ and therefore dimensionless.

**Figure 3 muscles-04-00029-f003:**
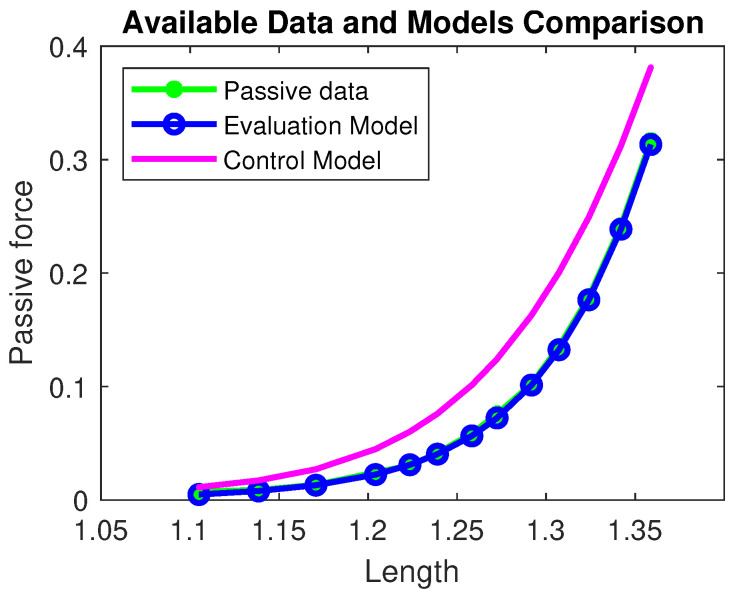
Passive force comparison among the available data (green), the Evaluation model (blue), and the Control model (magenta). The residual norm for the Evaluation model is $|r_{EM}{|}_{2}=0.0077$, while for the Control model it is $|r_{CM}{|}_{2}=0.1706$. All quantities are normalized with respect to the maximal active force $F_{o}$ and optimal length $L_{o}$ and therefore dimensionless.

**Figure 4 muscles-04-00029-f004:**
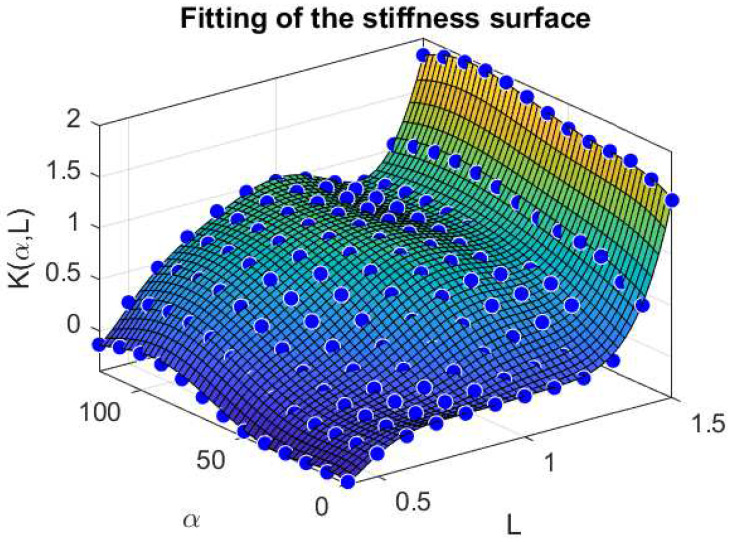
Fitting of the stiffness surface K(α,L) with respect to α and *L*. Force and Length quantities are normalized with respect to the maximal force $F_{o}$ and optimal length $L_{o}$, respectively, therefore, the resulting stiffness is dimensionless.

**Figure 5 muscles-04-00029-f005:**
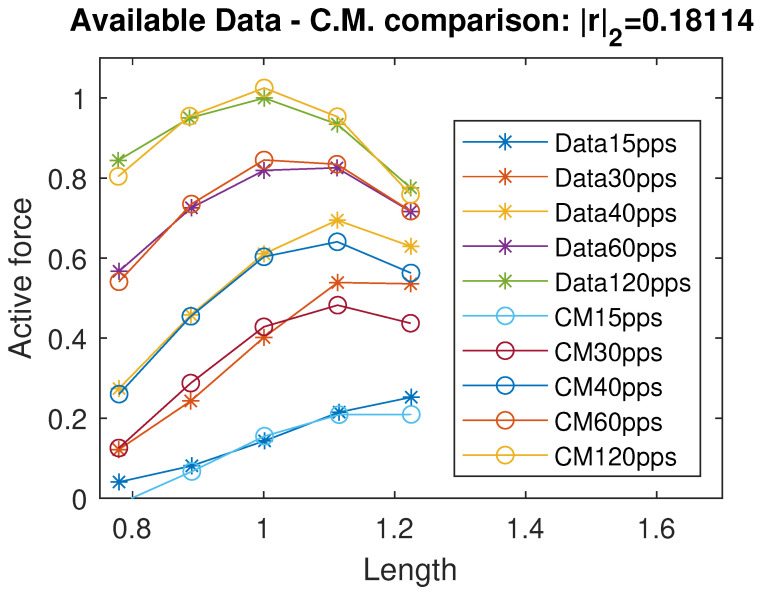
Comparison between the Control Model and experimental data. *r* denotes the residual vector. All quantities are normalized with respect to the maximal active force $F_{o}$ and optimal length $L_{o}$ and therefore dimensionless.

**Figure 6 muscles-04-00029-f006:**
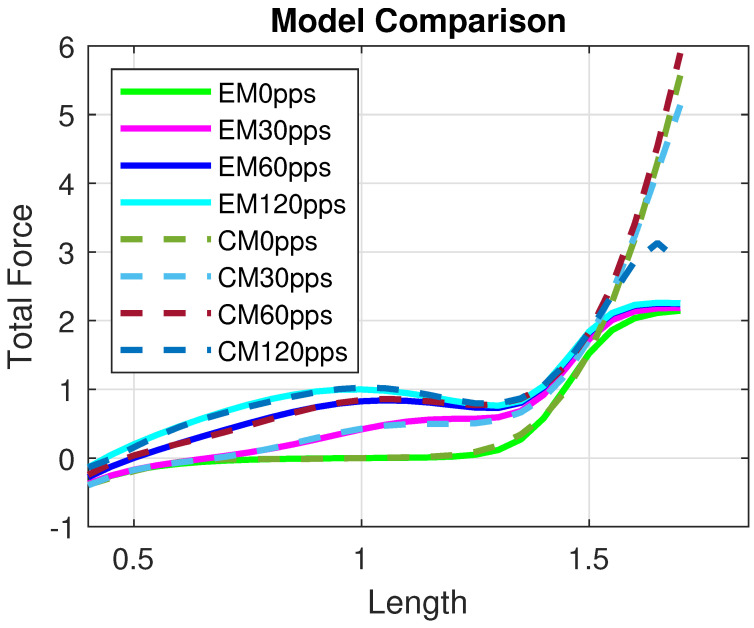
Comparison between the Evaluation Model and the Control Model for different excitation frequencies, i.e., α∈{0,30,60,120} pps. All quantities are normalized with respect to the maximal active force $F_{o}$ and optimal length $L_{o}$ and therefore dimensionless.

**Figure 7 muscles-04-00029-f007:**
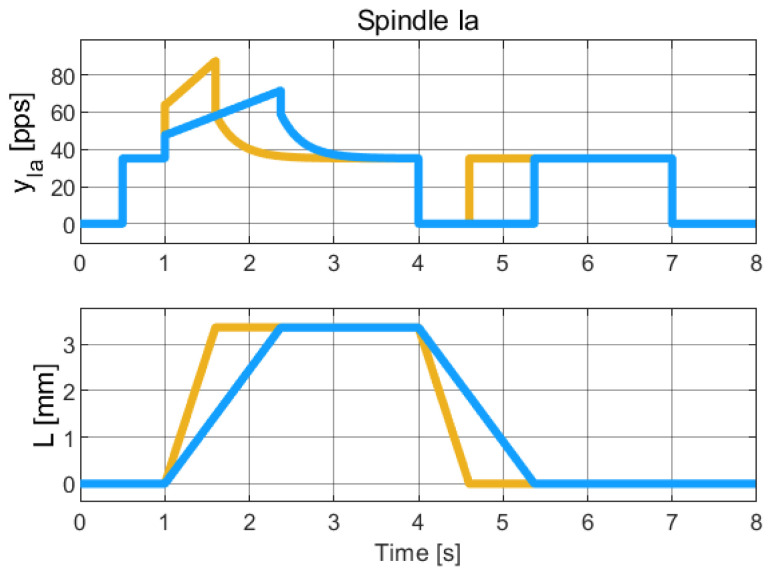
Simulation of the hybrid dynamical model of Spindle-Ia behavior, to be qualitatively compared with the experimental data reported in [22] (Chap. 19). The colors refer to different length profiles employed to collect the Spindle data Due to the limited resolution and quality of the published figures, reliable numerical extraction of the experimental data was not feasible. Therefore, this comparison is qualitative and intended to highlight the consistency in the overall trend and behavior of the signals.

**Figure 8 muscles-04-00029-f008:**
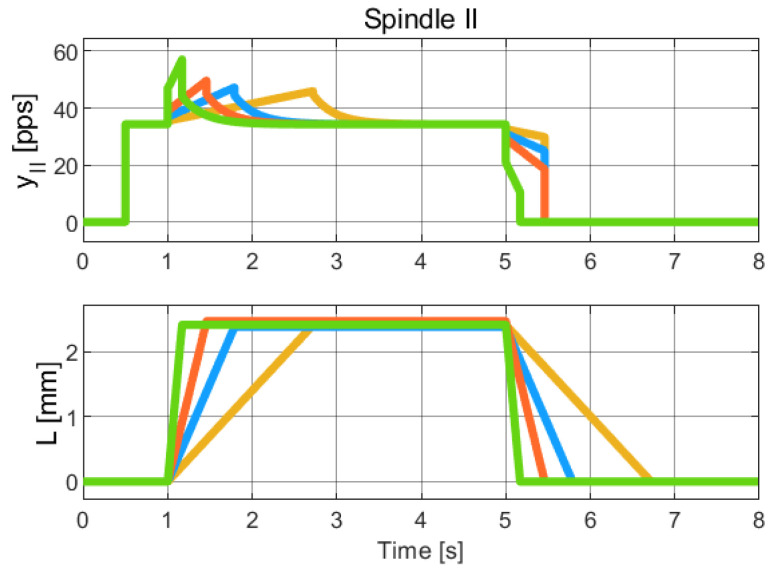
Simulation of the hybrid dynamical model of Spindle-II behavior, to be compared with the experimental data of Spindle-II in [22] (Chap. 19). The colors refer to different length profiles employed to collect the Spindle data. Due to the limited resolution and quality of the published figures, it was not feasible to obtain a reliable numerical extraction of the experimental data. Therefore, this comparison is qualitative and intended to highlight the consistency in the overall trend and behavior of the signals.

**Figure 9 muscles-04-00029-f009:**
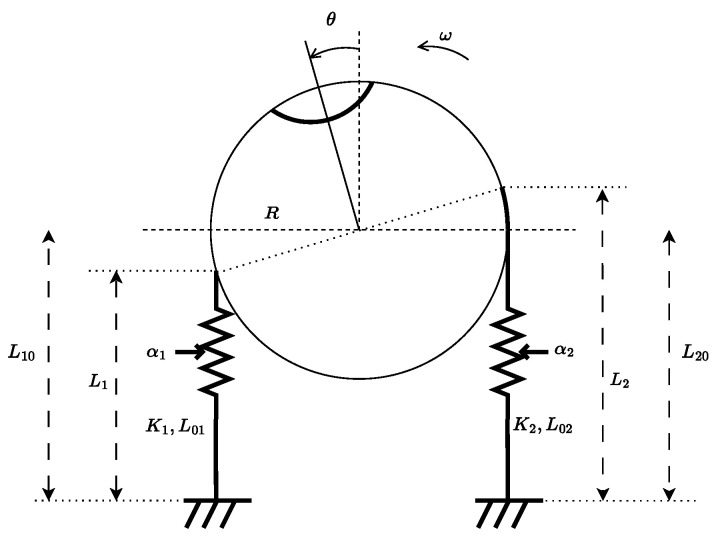
Schematic representation of the oculomotor system for horizontal plane motion. The system involves two antagonist muscles whose combined force generates a torque on the eyeball.

**Figure 10 muscles-04-00029-f010:**
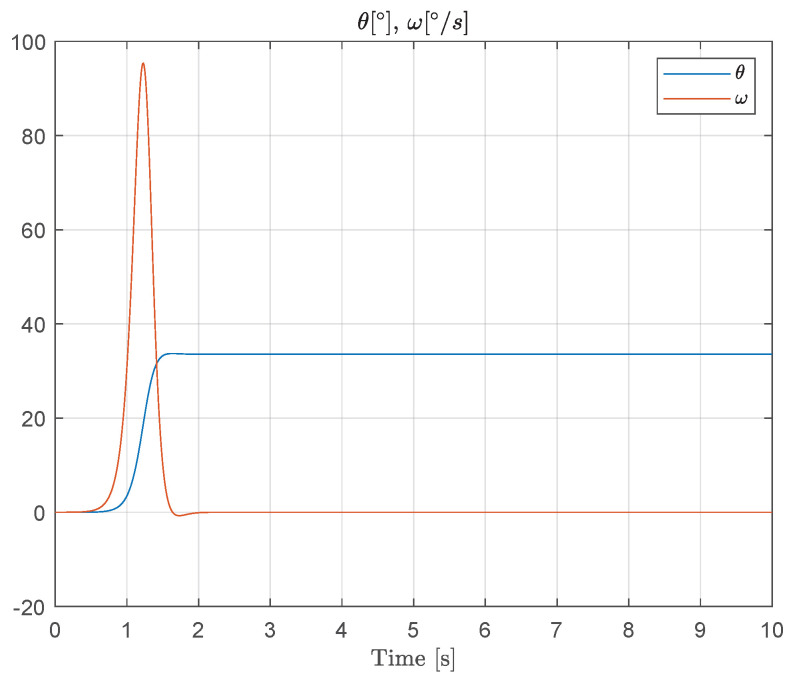
Simulated muscle force and spindle afferent response during a saccadic eye movement. Experimental data for this specific scenario are, to the best of our knowledge, not available in the literature. Therefore, the results presented are purely simulation-based and aim to illustrate the qualitative behavior predicted by the proposed model.

**Table 1 muscles-04-00029-t001:** Parameters of Evaluation Model active force. Residual value ${\left|r\right|}_{2}=0.050687$. The empty elements in the table are referred to as ’n.a.’, i.e., not available.

*i*	$c_{i1}$	$c_{i2}$	$c_{i3}$	$c_{i4}$	$c_{i5}$
1	1.134	0.723	0.0504	11.3297	643.259
2	0.574	−0.0211	0.8477	0.0008	n.a.
3	0.4518	−0.01704	−0.1235	0.04898	n.a.

**Table 2 muscles-04-00029-t002:** Parameters of Evaluation Model passive force. Residual value ${\left|r\right|}_{2}=0.0077$.

$f_{1}$	$f_{2}$	$f_{3}$	$f_{4}$	$f_{5}$	$f_{6}$
0.00381	2.16582	8.16634	1.45388	−7.32	−7.4

**Table 3 muscles-04-00029-t003:** Parameters of Control Model rest length.

$\ell_{1}$	$\ell_{2}$	$\ell_{3}$
$6.794\cdot 10^{-5}$	−0.01271	1

**Table 4 muscles-04-00029-t004:** Parameters of Control Model stiffness.

*i*	$k_{i1}$	$k_{i2}$	$k_{i3}$	$k_{i4}$	$k_{i5}$	$k_{i6}$	$k_{i7}$
1	−0.00041	0.0025	−0.0061	0.00763	−0.0052	0.0018	−0.00026
2	0.0586	−0.352	0.8444	−1.0277	0.6697	−0.2257	0.0319
3	−1.67504	9.829	−22.692	25.896	−15.268	4.5087	−0.5351
4	−40.977	244.666	−569.9335	664.1912	−405.645	120.853	−13.045

**Table 5 muscles-04-00029-t005:** Parameters of Spindle-Ia model.

${\bar{p}}_{\mathrm{Ia}}$	${\underline{p}}_{\mathrm{Ia}}$	${\bar{K}}_{L_{\mathrm{Ia}}}$	$K_{{\dot{L}}_{\mathrm{Ia}}}$	$K_{\gamma_{\mathrm{Ia}}}$
85	3.858	7.15	5.15	0.3514

**Table 6 muscles-04-00029-t006:** Parameters of Spindle-II model.

${\bar{p}}_{\mathrm{II}}$	${\underline{p}}_{\mathrm{II}}$	${\bar{K}}_{L_{\mathrm{II}}}$	$K_{{\dot{L}}_{\mathrm{II}}}$	$K_{\gamma_{\mathrm{II}}}$	$\bar{L}$
85	2.858	4.334	0.9184	0.343	2.5

**Table 7 muscles-04-00029-t007:** Model Parameters.

β	$L_{10}$	$L_{20}$	$\alpha_{1}$	$\alpha_{2}$
20	0.5	0.5	0.7	0.2

## Data Availability

No new data were created or analyzed in this study. Data sharing is not applicable to this article.

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
