# Peer review of "Modeling of Isometric Muscle Properties via Controllable Nonlinear Spring and Hybrid Model of Proprioceptive Receptors"

_muscles, 2025, doi:10.3390/muscles4030029_

Round 1
Reviewer 1 Report
Comments and Suggestions for Authors
The manuscript presents a thoughtful and technically rich modeling framework that brings a unique system-theoretic perspective to skeletal muscle function and sensor dynamics. While the approach is innovative and grounded in solid interdisciplinary theory, there remain several areas where the work would benefit from further refinement. Addressing these points mentioned below will significantly enhance the clarity, scope, and impact of the study.
Major Comments:
- The focus on isometric force is a good starting point, but the lack of modeling for dynamic muscle actions (like shortening or lengthening) limits how broadly the work can be applied. Since this is acknowledged as future work, it would be helpful to clearly mention this as a limitation in the discussion.
- The intermediate model is detailed but uses many parameters that don’t have clear physiological meaning. Even though the control model simplifies this, it still relies on complex equations. It might help to simplify further or explain the parameters in more physically meaningful terms.
- The models are built and validated using previously published data, but there's little discussion on how well these models would work across different types of muscles or in different species. Author should provide a broader validation to make the findings more robust.
- The sensor models simplify neural input by focusing mainly on one type of γ-motoneuron (γd). While this may be due to data constraints, the assumption should be better justified, especially since it might affect how well the model captures real feedback dynamics.
- The application to the oculomotor system is promising but feels incomplete. The critique of earlier models is valid, but without simulation or example results, the proposed improvements remain theoretical. Author should include a small study to strengthen this section.
Minor Comments:
- Several figures are labeled for “reviewer reference only.” If not meant for publication, author should remove or replace them. The manuscript has to be understandable for all the readers, not only for the reviewers.
- Terms like “rest length,” “optimal length,” and “Lo” vs. “L0” need consistent use and clearer definition to avoid confusion.
- The manuscript is well-referenced, though a few foundational works in modern neuromechanics (e.g., Zajac, 1989) might be included for broader context.
- Some equations have formatting inconsistencies or redundant brackets; a final pass for mathematical typesetting would improve readability.
- Terms like “rest length,” “optimal length,” and “Lo” vs. “L0” need consistent use and clearer definition to avoid confusion.
Reviewer 2 Report
Comments and Suggestions for Authors
While this work attempts to integrate control theory with biomechanical modeling, its scientific value is significantly undermined by three fundamental shortcomings: (1) the lack of demonstrable novelty beyond existing muscle models, (2) insufficient experimental validation relying solely on retrospective data fitting, and (3) critical omissions in addressing dynamic muscle behaviors essential for real-world applications. These limitations collectively diminish the study’s potential contributions. Below we detail the core issues necessitating rejection in its current form.
Lack of Attention to Basic Academic Standards
The author demonstrates a concerning lack of attention to detail, beginning with a glaring spelling mistake on the very first page—writing "Italian Istitute of Technology" instead of "Italian Institute of Technology." Such an error in the affiliation undermines the credibility of the work and suggests a careless approach to manuscript preparation. If the author neglects basic proofreading for their own institution’s name, it raises doubts about the rigor applied to the technical content, equations, and data analysis.
Sloppy Presentation and Low Scholarly Effort
The inclusion of placeholder figures (e.g., "This picture is shown here for the reviewer’s ease of reference, it will not be part of the final version") is unacceptable in a submitted manuscript. It indicates either haste or a lack of commitment to producing a polished, publication-ready work. Combined with the spelling errors and unaddressed copyright issues, these oversights suggest the author did not take the peer-review process seriously, warranting rejection on grounds of poor scholarship alone.
Unoriginal and Potentially Unethical Figure Use
Multiple figures (e.g., Figures 2–4) are directly reproduced from prior literature without substantial modification or novel analysis, serving only as "ease of reference for reviewers." Even more troubling, Figure 10 (b) is adapted from a published paper without a proper copyright declaration or permissions statement. This violates ethical publishing standards, as MDPI requires explicit permission for the reuse of artwork. The reliance on others' figures—without transformative interpretation or new data—further weakens the manuscript’s originality.
Failure to Address Pathophysiological Relevance and Research Gaps
Despite the extensive discussion of muscle modeling, the author completely overlooks the critical research gap in applying these models to understand pathophysiological conditions, such as muscle spasticity, atrophy, or neuromuscular disorders. The motivation section (Section 1.1) vaguely mentions "modeling pathological conditions affecting motor control" as a "long-term goal" but provides no concrete hypotheses, clinical correlations, or experimental pathways to achieve this. For instance, how could the proposed nonlinear spring model differentiate between healthy and dystrophic muscle properties? Could the hybrid spindle model simulate hyperreflexia in spasticity? Without addressing these questions, the work remains academically narrow and fails to justify its potential impact on biomedical research or therapeutic development.
Critical Points Justifying Rejection of the Manuscript
- Lack of Novelty and Incremental Contribution
- The proposed "controllable nonlinear spring" model is conceptually similar to prior work (e.g., [Winters, 1995]; [Brown et al., 1999]), with no clear theoretical or empirical advancement.
- The hybrid spindle models (Section 5) replicate known dynamics from [Mileusnic & Loeb, 2006] without significant innovation.
- Weak Experimental Validation
- Models are fitted to literature data only, with no new experiments or validation against independent datasets.
- The evaluation model’s residuals (e.g., ∣r∣2 = 0.18114 for the control model) suggest poor predictive accuracy, especially for L > 1.5L0​.
- Over-Simplified Assumptions
- The stiffness function K(α,L) (Eq. 13) uses a high-order polynomial (6th degree) without biomechanical justification, risking overfitting.
- The oculomotor application (Section 6) ignores time delays in neural feedback, a critical flaw given known delays in spindle responses ([Windhorst, 2007]).
- Incomplete Dynamic Modeling
- Focuses solely on isometric force (FL+Fp​), neglecting eccentric/concentric dynamics (FV​), which are essential for real-world movement.
- No discussion of fatigue, hysteresis, or tendon compliance, which are key in muscle physiology ([Lieber, 2002]).
- Poor Clarity and Presentation
- Figures 2–4 are placeholder sketches with no final versions provided.
- Table 4’s stiffness parameters lack units or physiological interpretation.
- The oculomotor section (Section 6) is redundant, rehashing torque redundancy without new insights.
- Ignored Contemporary Literature
- Fails to engage with recent advances in muscle synergies ([d’Avella et al., 2006]) or reflex modulation ([Pruszynski et al., 2016]).
- No comparison to computational muscle models (e.g., OpenSim’s Millard model) used in biomechanics.
- Questionable Practical Utility
- The control model’s complexity (12 parameters for L0(α), 28 for K(α,L)makes it unusable for real-time applications (e.g., prosthetics).
- No discussion of scalability to multi-muscle systems or closed-loop stability.
Reviewer 3 Report
Comments and Suggestions for Authors
Excellent article. It offers a new model to explain isometric force generation
Author Response
Comment 1: Excellent article. It offers a new model to explain isometric force generation
Response 1: We sincerely thank the reviewer for the positive feedback and appreciation of our work. We are glad that the proposed model to explain isometric force generation was found valuable.
Round 2
Reviewer 1 Report
Comments and Suggestions for Authors
The revised manuscript demonstrates a substantial improvement over the previous version. Author has carefully addressed all reviewer comments. Key enhancements include clearer articulation of model assumptions and limitations, improved interpretability of parameters, consistent terminology, especially regarding rest and optimal muscle lengths, and the inclusion of a simulation study to support the oculomotor system application. The figures and mathematical expressions have also been refined for clarity. Overall, the manuscript is now well-structured and scientifically sound, and I consider it ready for publication.
Author Response
Comment: The revised manuscript demonstrates a substantial improvement over the previous version. Author has carefully addressed all reviewer comments. Key enhancements include clearer articulation of model assumptions and limitations, improved interpretability of parameters, consistent terminology, especially regarding rest and optimal muscle lengths, and the inclusion of a simulation study to support the oculomotor system application. The figures and mathematical expressions have also been refined for clarity. Overall, the manuscript is now well-structured and scientifically sound, and I consider it ready for publication.
Response: Thank you very much for your thoughtful and encouraging feedback. We sincerely appreciate the time and effort you devoted to reviewing my manuscript, as well as your constructive comments throughout the revision process.
We are pleased to know that the improvements made in the revised version have addressed your concerns and that you now consider the manuscript ready for publication.
Best regards,
Mario Spirito
Reviewer 2 Report
Comments and Suggestions for Authors
While Version 2 shows superficial improvements in presentation and structure, it fails to address the core scientific, ethical, and translational issues raised in the initial review. Without original data, ethical compliance, and meaningful biomedical relevance, the manuscript remains unsuitable for publication.
Revision needed:
- Figure 1: This figure is noted as being adapted from a reference. Please clarify whether it is directly copied or redrawn. If it is directly copied, appropriate permission must be obtained, especially if there are copyright restrictions. If it has been redrawn, kindly specify the software used to create the image (e.g., BioRender, Illustrator, etc.).
- Figure 2: This figure presents a data comparison with the dataset provided in reference [12], which is appropriate. However, in Figures 7 and 8, the caption only states that the results are “to be compared with the experimental data of Spindle-Ia in [22][Ch.19].” This phrase is vague. For proper validation of the simulation results, the corresponding experimental data should be included in the figure. Ideally, the authors should extract the numerical data from the experimental figure (e.g., from Version 1 of the manuscript), overlay it on the simulation results, and assess the match. If the units differ between the simulation and the experiment, normalization can be applied to enable a meaningful comparison.
- Figure 10: Is there any experimental data available to validate the simulation results presented in this figure? If so, it should be included or referenced clearly.
- Figure Axes and Units: Several figures are missing units on the x and y axes. For instance, in Figure 5, the x-axis indicates “length” but no unit is provided. All axes should be properly labeled with both quantity and units to ensure clarity and reproducibility.
- Formatting Suggestion: Many paragraphs throughout the manuscript are only 2–3 lines long. These could be combined to form more cohesive, logically structured paragraphs, which would improve the overall readability and flow of the text.
- The control model retains excessive complexity (e.g., 12 parameters for Lâ‚€(α), 28 for K(α,L)), making it impractical for real-time applications like prosthetics or robotics. There is no discussion of scalability to multi-muscle systems or closed-loop control, which limits translational potential.
- The core modeling framework is unchanged: high-order polynomial stiffness functions (e.g., Eq. 13) are still used without biomechanical justification, risking overfitting and poor generalizability. Validation remains limited to curve-fitting against literature data. No new experiments or independent datasets are introduced, and predictive accuracy is weak—especially for muscle lengths beyond 1.5Lâ‚€.
Round 3
Reviewer 2 Report
Comments and Suggestions for Authors
In the third revised MS, the figure captions state that all quantities are normalised and therefore dimensionless. Typically, normalising implies that both the X and Y axes range from 0 to 1. However, in some figures—such as Figure 6—the Y axis ranges from -1 to 6, and the X axis from 0.5 to 1.5. This raises a question about the consistency of normalisation and its interpretation within these plots.
Author Response
Comment : In the third revised MS, the figure captions state that all quantities are normalised and therefore dimensionless. Typically, normalising implies that both the X and Y axes range from 0 to 1. However, in some figures—such as Figure 6—the Y axis ranges from -1 to 6, and the X axis from 0.5 to 1.5. This raises a question about the consistency of normalisation and its interpretation within these plots.
Response :
We thank the reviewer for the observation regarding normalization in the figures. As clarified in the captions of the revised manuscript, all quantities are normalized with respect to the maximal active force F_o and the optimal length L_o, and are therefore dimensionless.
This normalization choice was introduced and explained in Section 2 (lines 150–157) of the manuscript. Consequently, the ranges observed in the figures (e.g., 0.5 to 1.5 for length, or force values exceeding 1) are consistent with known muscle behavior relative to L_o and F_o. For example, force can exceed F_o when passive and active components are summed, and lengths can vary around L_o, depending on muscle stretch or contraction.
We hope this clarifies the consistency and interpretation of the normalization used throughout the manuscript.
Sincerely,
Mario Spirito